**METHOD**

# DREAMS: deep read-level error model for sequencing data applied to low-frequency variant calling and circulating tumor DNA detection

Mikkel H. Christensen[1,2†], Simon O. Drue[1†], Mads H. Rasmussen[1,2†], Amanda Frydendahl[1,2†], Iben Lyskjær[1,2], Christina Demuth[1], Jesper Nors[1,2], Kåre A. Gotschalck[2,3], Lene H. Iversen[2,4], Claus L. Andersen[1,2*†] and Jakob Skou Pedersen[1,2,5*†]

†Mikkel H. Christensen, Simon O. Drue, Mads H. Rasmussen, and Amanda Frydendahl shared the first authorship.

†Claus L. Andersen and Jakob Skou Pedersen shared the senior authorship.

*Correspondence:
cla@clin.au.dk; jakob.skou@clin.au.dk

¹ Department of Molecular Medicine, Aarhus University Hospital, Aarhus, Denmark
² Department of Clinical Medicine, Faculty of Health, Aarhus University, Aarhus, Denmark
³ Department of Surgery, Horsens Regional Hospital, Horsens, Denmark
⁴ Department of Surgery, Aarhus University Hospital, Aarhus, Denmark
⁵ Bioinformatics Research Center, Faculty of Science, Aarhus University, Aarhus, Denmark

## Abstract

Circulating tumor DNA detection using next-generation sequencing (NGS) data of plasma DNA is promising for cancer identification and characterization. However, the tumor signal in the blood is often low and difficult to distinguish from errors. We present DREAMS (Deep Read-level Modelling of Sequencing-errors) for estimating error rates of individual read positions. Using DREAMS, we develop statistical methods for variant calling (DREAMS-vc) and cancer detection (DREAMS-cc). For evaluation, we generate deep targeted NGS data of matching tumor and plasma DNA from 85 colorectal cancer patients. The DREAMS approach performs better than state-of-the-art methods for variant calling and cancer detection.

**Keywords:** Circulating tumor DNA, Next-generation sequencing, Cancer research, Colorectal cancer, Machine learning

## Background

Degraded DNA fragments are released into the blood through apoptosis, necrosis, and active secretion from a range of cell types and can be detected as circulating free DNA (cfDNA) [1]. Solid tumors also shed DNA into the bloodstream and cfDNA of cancer origin is called circulating tumor DNA (ctDNA) [2]. The ctDNA level in blood is reported to be positively associated with tumor burden [3, 4]. As the half-life of cfDNA is less than an hour, ctDNA measurements can be considered real-time assessments of tumor burden and studies have shown that ctDNA can be more sensitive than radiological imaging [5–7]. This makes ctDNA measurements a promising approach for detecting relapse in patients who have undergone curative surgery [6–10]. Other proposed applications include diagnosis and intervention planning,

**Fig. 1** Error generation in next-generation sequencing data. Normal cells (gray) and cancer cells (blue) shed DNA into the bloodstream. The cancer DNA (blue) contains a mutation (yellow star). The circulating free DNA in the blood is damaged both in vivo and in vitro (green triangle). Errors can be introduced at each PCR duplication during amplification (red circle). Further errors are accumulated during sequencing and mapping (purple square). The final data contains mapped reads, where some mismatches are errors, and others are mutations from tumor cells

tracking therapeutic response, monitoring the development of treatment resistance, and ultimately early detection of cancer in screening programs [8, 11]. Since obtaining liquid biopsies, such as plasma from blood samples, is both cost-effective and minimally invasive, techniques for efficient ctDNA detection hold great promise for targeted treatment in precision medicine.

In clinical contexts with low tumor burden, e.g., detection of minimal residual disease after curative-intended surgery and early detection of recurrence, the ctDNA constitute only a minor fraction of the cfDNA, often less than 0.1%. Hence, the error rate of current sequencing methods is in the same order of magnitude as the tumor signal [12], making it challenging to accurately distinguish errors from true mutations in ctDNA applications. Errors can arise in several steps between the initial shedding of cfDNA and the final generation of next-generation sequencing (NGS) reads (Fig. 1). DNA fragments may be damaged, e.g., by deamination or oxidation [13, 14], during PCR amplification of the sequencing library [13], and during sequencing from PCR amplification and/or sequencing artifacts [13]. For deep sequencing, some of the PCR and sequencing errors can be rectified using unique molecular identifiers (UMIs). With the use of UMIs, each DNA fragment is labeled with a unique "barcode" prior to PCR amplification, such that replicates of the same fragment can be grouped together. Errors can then be eliminated by comparing the replicates within a group, as errors from PCR amplification and sequencing are likely to be present in only a minority of reads. However, some errors, such as DNA damage introduced prior to UMI labeling, remain and continue to challenge the discrimination of true low-frequency mutational signal from these errors.

Several methods for detecting low-frequency variants using NGS data have been developed. Most of these establish a model for the expected frequency of errors and

then assess the mutational signal with a statistical test. They differ greatly in the required data prerequisites, how the errors are modeled and handled, and the final assessment of the mutational signal.

Mutect2 [15], Strelka2 [16], and Shearwater [17] are examples of general somatic variant callers applicable for most NGS data. Mutect2 realigns reads in regions with mutational signal and then calculates a log-odds for the existence of the alternative allele using a statistical model in which the error rates are derived from the PHRED scores. Similar to Mutect2, Strelka2 realigns the reads and then uses a statistical model to determine the likelihood of a variation being real by analyzing base quality, read mapping quality, and depth at each position. Shearwater is developed specifically for low-frequency somatic variant detection for sub-clonal tumor mutations. It builds a position-specific error model based on the observed rate of read alignment mismatches across a set of training samples. A mutation is called if the observed signal exceeds what is expected from the error model. Additionally, this method can incorporate prior knowledge about the probability of the mutations of interest.

Other methods, including MRDetect [18] and INVAR [19], have been specifically tailored to detect ctDNA in NGS data. These methods build on the idea of aggregating the signal across multiple mutations to classify a sample as ctDNA positive or negative, as opposed to calling each individual mutation. For this purpose, a patient-specific catalog of mutations is generated from a matched tumor sample. However, the enhanced performance of these methods comes at the expense of general applicability as they assume the presence of curated data from known ctDNA fragments or specialized lab protocols. Another approach, iDES [12], finds mutations in paired reads by combining a specialized stranded barcoding scheme and a polisher that aims to filter out an erroneous mutational signal based on a number of criteria, including an error model. Mutations are then called based on the remaining variant signal.

Here we develop a generally applicable ctDNA detection method based on a detailed background error model of individual read positions. This approach aims to capture general read-level error behavior and thus be applicable even for genomic regions where training data is not available. Data from reads known to come from ctDNA is not needed, and all data outside known mutated positions, or from independent normal samples can be used as training data. However, training data that was obtained similarly to the test data will provide the most precise model. Thus, severe changes in laboratory protocols should optimally be accompanied by re-training the model. Some features such as the read position [20], proximity to fragment ends [14], UMI group size [12], GC-content [21], and trinucleotide context [22] have been shown to affect the probability of errors at individual read positions. By modeling their combined effect, the error rate of individual read positions may be predicted. Thereby, a read alignment mismatch, i.e., a non-reference base, with a low predicted error rate can provide more mutational evidence than a mismatch with a high error rate. This allows for improved cfDNA error modeling, which is key to develop accurate ctDNA applications.

In the following, we demonstrate how cfDNA errors can be modeled accurately using a neural network, by combining read-level features with information about the sequencing context. For this we developed DREAMS (Deep Read-level Modelling of Sequencing-errors) that incorporates both read-level and local sequence-context features for

positional error rate estimation. Based on DREAMS, we developed a method for variant calling (DREAMS-*vc*) to accurately call individual cancer mutations in cfDNA data. The method was generalized for cancer calling in DREAMS-*cc* that aggregates the signal across a catalog of mutations for accurate estimation of the tumor fraction and sensitive determination of the overall cancer status. To evaluate the performance of DREAMS, we performed deep-targeted sequencing of pre- and postoperative cfDNA samples from 85 stage I–II colorectal cancer (CRC) patients and compared to state-of-the-art methods Mutect2 [15] and Shearwater [17].

## Results

Plasma cfDNA was extracted from preoperative (Pre-OP) and postoperative (Post-OP) blood draws of 85 stage I-II CRC patients (Table 1) undergoing curative surgery. In addition, data from two stage III CRC patients were used in the model training. A biopsy from the resected tumor and paired peripheral blood cells was sequenced to generate a patient-specific mutational catalog. Post-OP samples were collected 2–4 weeks after surgical removal of the primary tumor (Fig. 2). Each cfDNA sample was sequenced using a custom hybrid-capture panel, designed to capture 41 exonic regions, spanning 15.413 bp, frequently mutated in CRC (Additional file 1: Section S1 and Additional file 1: Table S1). After collapsing UMI groups, the median of the average depths with corresponding interquartile range (IQR) of samples were for Pre-OP, 3307 (IQR: 3560); Post-OP, 7143 (IQR: 8844); buffycoat, 1850 (IQR: 1468); and tumor samples, 2132 (IQR: 2145); no samples had an average read depth below 100. All samples have been mapped and processed through the same pipeline (Additional file 1: Section S1).

We first identified features that are known or expected to affect the error rate (Fig. 3a). In general, they can be split into two types: local sequence-context features

**Table 1** Clinical characteristics

| Characteristic | Count or median (percent or range) |
|---|---|
| Patients | 85 (100%) |
| Gender | |
| *Male* | 53 (62%) |
| *Female* | 32 (38%) |
| Age [years] | 71 (49–87) |
| Tumor location | |
| *Right colon* | 23 (27%) |
| *Left colon* | 26 (31%) |
| *Rectum* | 36 (42%) |
| Pathological T-stage | |
| *pT1* | 15 (18%) |
| *pT2* | 25 (29%) |
| *pT3* | 41 (48%) |
| *pT4* | 4 (4.7%) |
| UICC stage | |
| *I* | 40 (47%) |
| *II* | 45 (53%) |

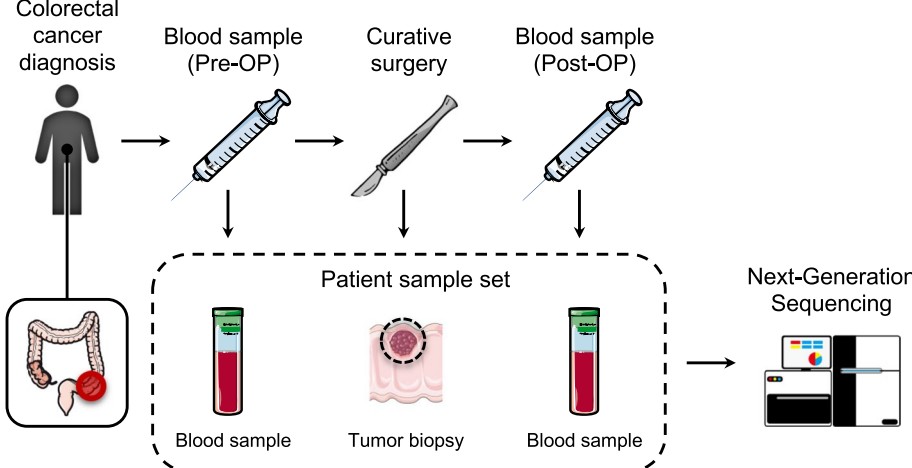

**Fig. 2** The data collection setup for tumor-informed relapse detection in colon cancer patients. After the patient is diagnosed, a liquid biopsy is extracted prior to curative surgery (Pre-OP). A biopsy is taken from the resected tumor. Following surgery liquid biopsies (Post-OP) can be collected to monitor relapse. All collected samples are sequenced using next-generation sequencing

and read-level features. The local sequence-context features capture the genomic sequence context, including the trinucleotide context, information about the sequence complexity (Shannon entropy of nucleotide, 1-mer, or dinucleotide, 2-mer, frequency), and GC contents in an 11 bp window around the position of interest (see the "Methods" section).

The read-level features capture the structural composition of the read, UMI characteristics, and sequencing information. The structural composition includes the strand a read aligns to (forward or reverse), the number of insertions and deletions in the read, and the total size of the underlying fragment. In the read pre-processing, UMIs were used to generate consensus reads with lowered error rates (Additional file 1: Section S2). For each consensus read, we extracted the UMI-group size, the number of reads disagreeing with the consensus at the position, and the overall number of mismatches outside the position of interest. As sequencing-related features, we included the base position in the read (read position), the length of the read sequence after overlap trimming, and whether the read is the first to be sequenced from the read-pair. The read quality (PHRED score) was not included, as it had the same high value for all positions in the UMI-collapsed consensus reads.

We evaluated the individual feature associated with the error rate by analyzing the total set of read alignment mismatches ($n = 707,562$) across all Post-OP samples (Fig. 3b–d), after excluding mutations and variants found in matching tumor and germline samples. The mismatches were compared to an equal number of randomly sampled matches, to estimate the error rate for each feature across its values (Additional file 1: Section S3).

Since fragment lengths of cfDNA are influenced by nucleosome binding patterns, the fragment length distribution has peaks at around 162 bp (mono-nucleosomal) and 340 bp (di-nucleosomal) [23]. The error rate tended to be minimized in fragments of these lengths (Fig. 3b). As expected, we observed a lower error rate in consensus reads formed by larger UMI groups [12] (Fig. 3c).

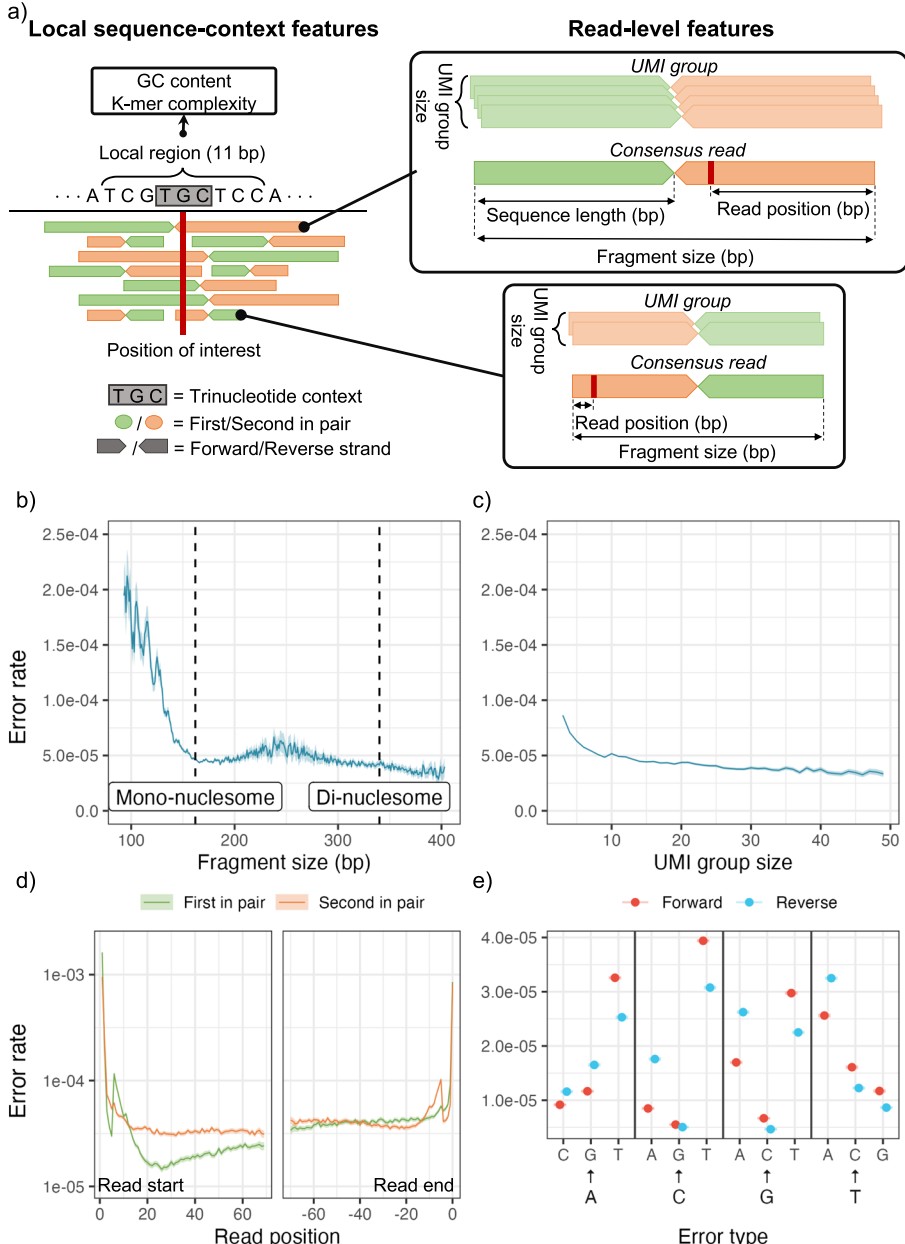

**Fig. 3 a** Examples of local sequence-context features and read-level features extracted from reads covering a position of interest in a read mapping. Centered at the position of interest, the trinucleotide context is extracted, and the surrounding 11-bp region is used for calculating regional features, including GC content and K-mer complexity. The read pairs contain a forward and reverse read that are enumerated as either the first or second of the pair according to the order of sequencing. Two read pairs are used for illustration of the read-centric features in the panels on the right. The UMI groups are shown to indicate the variation in the number of reads used for the consensus reads. The read position and fragment size are shown for the consensus reads. **b–e** Variation in observed error rate for selected features based on their observed distribution: **b** fragment size, **c** UMI group size, **d** read position relative to the start and end of reads and the variation between the first and second read in a pair. **e** Error type for each strand (forward and reverse). For each feature, the 95% confidence interval is indicated by the shaded areas or error bars. See Additional file 1: Section S3 for how the error rates and confidence intervals are calculated and similar plots of the remaining features

The error distribution for the read position showed an increased error rate in the beginning and end of the reads (Fig. 3d). We also observed a clear difference in error distribution along the read between the first and second read of the pair. The 12 different nucleotide alterations showed widely different error rates (Fig. 3e), which is expected as error-induced mismatches are not equally likely, and the rate further differed between the two strands. However, strand symmetric alterations were generally similar, apart from the mismatches C➔T/G➔A and C➔A/G➔T.

Overall, we saw variation in the error rate for all the presented features (the remaining are shown in Additional file 1: Section S3). Thus, for a given genomic position, different reads may have different error rates due to differences in read-level features. In the following, we present how this variation can be captured and used to potentially improve detection of ctDNA.

### Neural network model and feature selection

To predict the error rate at a given read position, we used a neural network model with the input features described above (see the "Methods" section). The predictive ability of individual features was evaluated using a "leave-one-covariate-out" (LOCO) scheme [24] (Additional file 1: Section S4). In short, we evaluated the performance of a full model containing all features (baseline) and then the relative performances of restricted models where each feature had been left out one by one. We used the latter to measure and rank the importance of each feature (Fig. 4a). When leaving out the trinucleotide context, the reference base was provided instead to assess only the importance of the two neighboring nucleotides.

We found the most and second most informative feature for modeling the error rate to be respectively the read position and the strand (Fig. 4a). The third feature was the trinucleotide context, indicating that there is a difference in error rate for different contexts, as found by others [19]. The fourth feature was whether the read was the first in the read pair to be sequenced, indicating that systematic errors might be induced by the lab protocol. The fragment length, sequence length, and UMI group size also contribute significantly to the model. The remaining features showed little to no effect on the model performance. This showed that read-level features do contribute to accurate modeling of

(See figure on next page.)
**Fig. 4** **a** Features are individually removed one-by-one from the full model containing all features to measure the decrease in validation error. The most important feature is then defined as the one that if removed decreases the validation error the most, and vice versa. The gray points show the mean decrease in validation error for each fold of a fivefold cross-validation. The average of these is used to rank the features by importance, indicated by the black points. **b** Based on the importance ranking, the features are cumulatively removed one-by-one from a full model. If the decrease in validation error compared to the full model is significant, the feature should not be removed from the model. A feature is only kept if removing it worsens the performance in all folds of the fivefold cross-validation. **c** Structure of the neural network model. The neural network uses three different types of input features: numeric, categorical, and embedded. The input features are processed differently in each group. The input features are then parsed through three hidden layers of decreasing width. The output contains 4 nodes representing the probability of observing each of the four based (A, T, C, G) at the given read position. **d** Illustration of 5 × 2-cross-validation procedure for the estimation of performance. The patients are first split into two approximately equally sized folds. The neural network model is trained on the Post-OP data of fold 1 and validated by testing the models on the Pre-OP samples of the other fold. This is then repeated by swapping the data in fold 1 and 2. The whole process is repeated 5 times

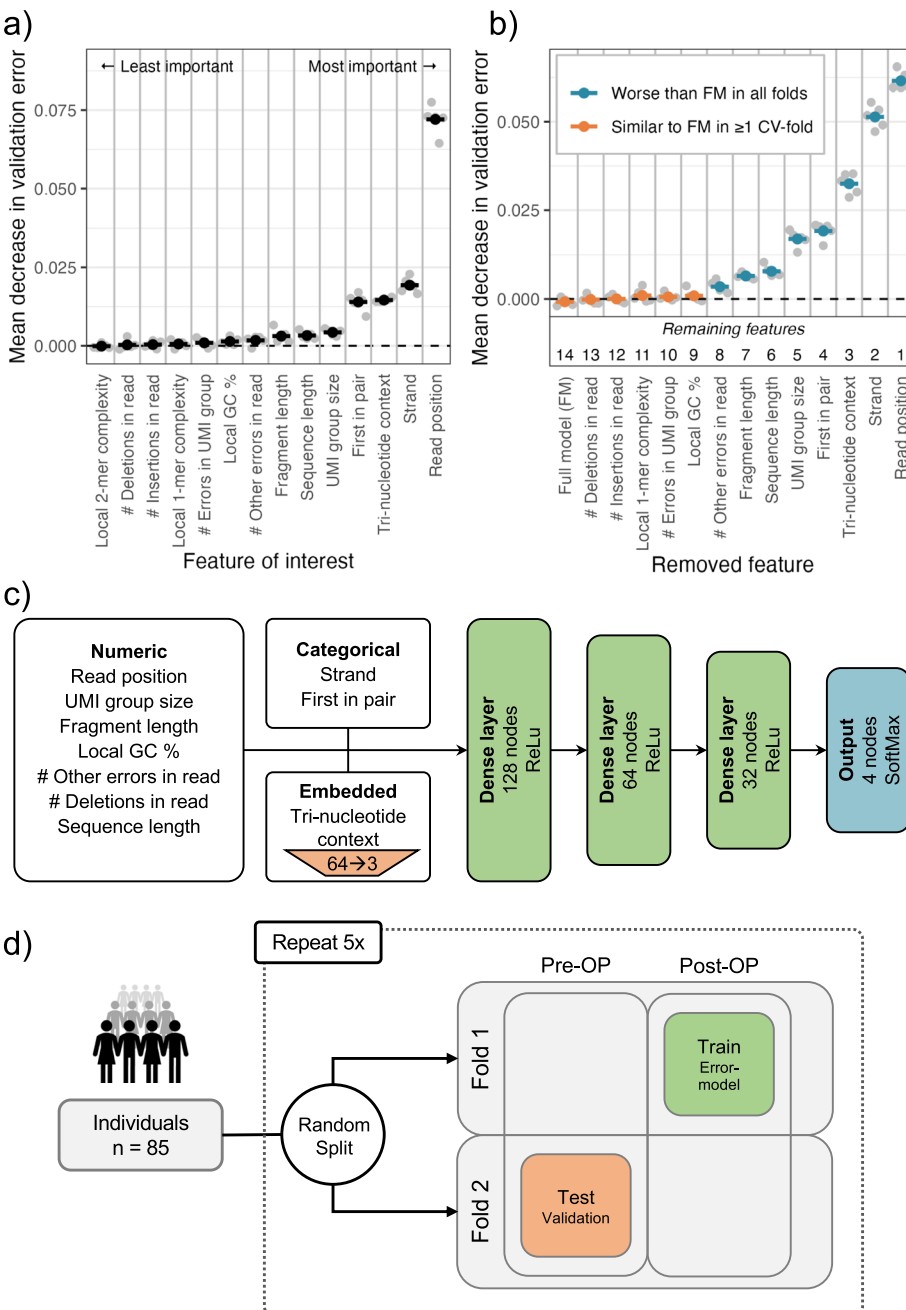

**Fig. 4** (See legend on previous page.)

the error rate, and that they might be at least as important as features derived from the local sequence context.

An optimal subset of informative features was chosen using a stepwise procedure where features were excluded in order of importance (see the "Methods" section). The set of features chosen was the smallest model that did not perform significantly worse than the full model (Additional file 1: Section S4). The five least important features could be removed without any significant negative effect on the performance (Fig. 4b). Of the remaining eight features, seven were read-level features, namely the number of other

errors in the read, the fragment length, the sequence length, the UMI group size, and if the read was first in pair, strand, and read position. The only remaining local sequence-context feature was the tri-nucleotide context.

The numerical and categorical variables are processed differently in the neural network prior to the hidden layers (Fig. 4c). The numerical features are batch normalized, the categorical features are one-hot encoded, and the tri-nucleotide context is embedded in three dimensions to handle the large number of possible contexts (see the "Methods" section).

To validate the utilization of the DREAMS error model, we applied it in calling tumor variants (DREAMS-*vc*) and cancer (DREAMS-*cc*) (see the "Methods" section). We assessed the performance using five repeats of twofold cross-validation ($5 \times 2$ CV) (Fig. 4d). The model was trained on the Post-OP samples, and Pre-OP samples were used for method validation. The tumor variant positions were excluded from the training data. Potential signals from sub-clonal variants that are not detected in the tumor should be infrequent and present at low levels and therefore have minimal effect on the error model. By excluding variants found in the germline samples, we expect to reduce the potential signal of clonal hematopoiesis of indetermined potential (CHIP). The split was done on patient level to ensure that a model is not trained and tested on data from the same patient. This analysis was repeated with five different randomized splits to control for split-induced variation.

### Predictive performance in clinical data

The performance of calling tumor mutations in the plasma samples was assessed by looking at the area under the receiver operating characteristic curves (AUC). The performance of DREAMS-*vc* was compared to state-of-the-art algorithms Mutect2 and Shearwater. All positions with at least one observed mismatch were scored with each method and included in the performance calculations (Fig. 5a). Positions without signal was called negative by any method, making them redundant for performance comparisons. DREAMS was also compared to iDES [12] and Strelka2 [16]; however, these methods are not directly comparable and are therefore presented in Additional file 1: Section S5. In addition, we evaluated how the performance of DREAMS depended on the size of the training set and found it increased with the number of training samples, with the largest gain seen when going from one to two samples (Additional file 1: Section S6).

Using DREAMS-*vc*, we aimed to call the tumor mutations of each patient from their respective mutation catalog. As negative controls, we attempted to call cross-patient mutations, by searching for the mutations found in other patients. Additionally, within the sequencing panel, a set of 500 randomly selected positions each associated with a random alteration, was used as negative controls, and referred to as validation alterations.

Evaluating across the combined negative set of both cross-patient mutations and validation alterations and cancer stages, DREAMS-*vc* performs significantly better than both Shearwater and Mutect2 (Fig. 5a). Additionally, the performance was assessed separately for stage I and stage II CRC patients. This showed superior performance of DREAMS-*vc* especially for the stage II CRC patients (Fig. 5a). As expected, all models perform better on later-stage patient samples as these are expected to have a higher mutational signal in

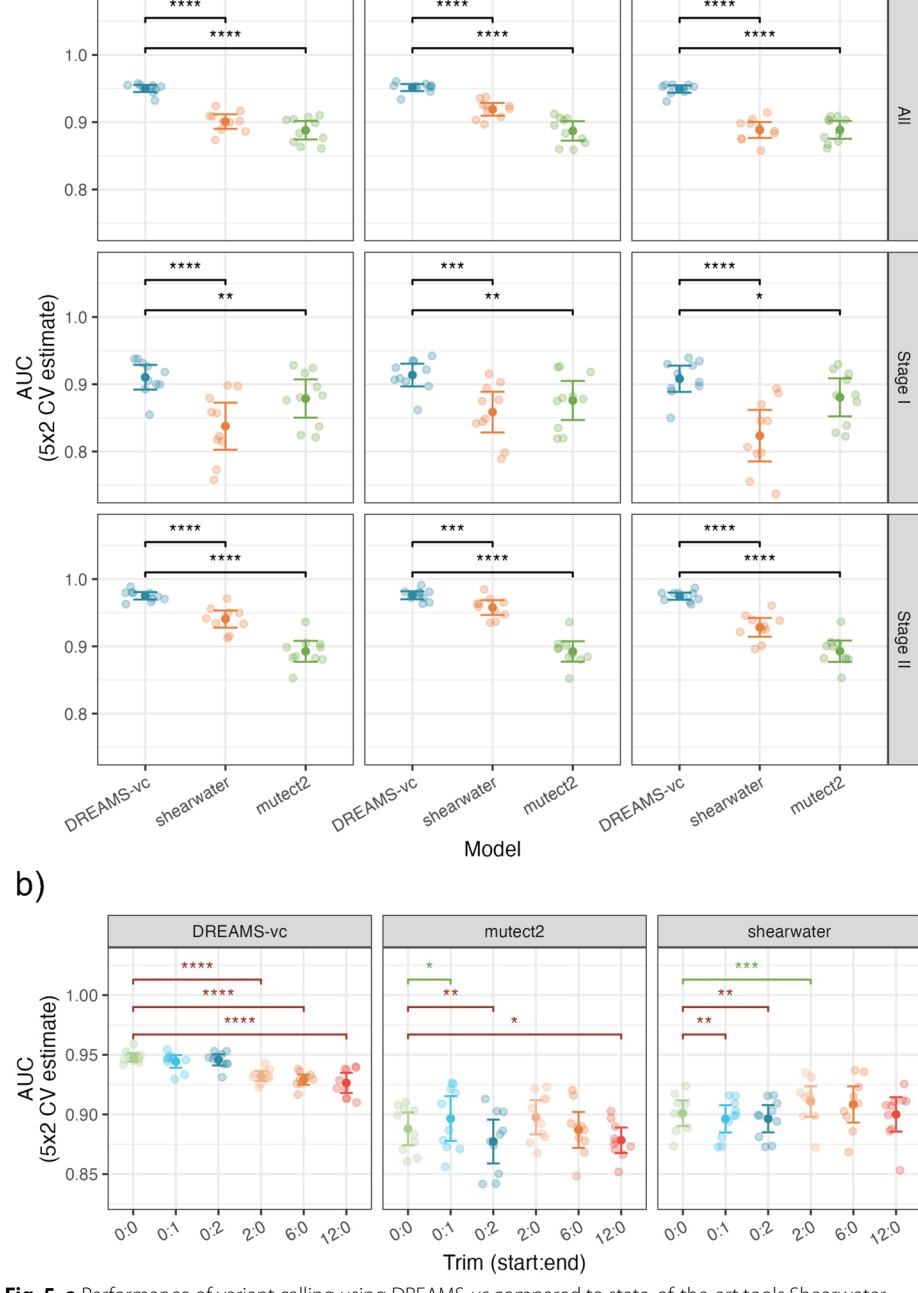

**Fig. 5** **a** Performance of variant calling using DREAMS-*vc* compared to state-of-the-art tools Shearwater and Mutect2. The AUC is estimated based on the different negative sets: The cross-patient calls, 500 random validation alterations, and these sets combined (All). The AUC is also estimated for the full group of patients (All), and the patients with stage I and stage II CRC, individually (ns: $p \geq 0.05$, *: $p < 0.05$, **: $p < 0.01$, ***: $p < 0.001$, ****: $p < 0.0001$). **b** Performance of variant calling using DREAMS-*vc*, Shearwater, and Mutect2 with various trimmings of the reads (start:end). The performance is compared pair-wise to the 0:0 model, with worse (red) and improved (green) performances indicated

the cfDNA due to a higher tumor burden. The methods perform more similarly on stage I patients; however, DREAMS-*vc* has significantly better performance.

Performance evaluations for each of the separate negative sets showed that DREAMS performs better than Mutect2 with the cross-patient negative set and better than Shearwater with the validation set as the negative set. The variation in performance of DREAMS-*vc* across splits and folds is lower than for Mutect2 and Shearwater, which indicates that its variant calling is more stable across patients and mutation types.

The explorative feature analysis indicated an increased error rate at the beginning and end of reads (Fig. 3d). DREAMS takes the read position into account and can thereby compensate for the increased error rate at these positions. Other methods such as Mutect2 and Shearwater are not aware of read ends and the performance of these can potentially improve by trimming these. To investigate the effect of trimming, we evaluated the performance of each method when trimming 0, 2, 6, or 12 of the bases in the beginning of reads or 0, 1, or 2 of the bases in the end of reads (Fig. 5b). We found that Shearwater can improve performance by trimming 2 bases from the beginning of reads and Mutect2 can improve by trimming the last base of each read. For DREAMS, the performance is only decreased when trimming, especially in the beginning of reads.

By setting the false-positive rate at 5% for each method across the validation alterations with any support, we get comparable thresholds for the three confidence measures: *p*-values, Bayes factor, and TLOD for DREAMS-*vc*, Shearwater, and Mutect2, respectively. This allows for a comparison of the sensitivity of the models at a pre-determined specificity of 95%. The methods could then be assessed across an alteration catalog of 191 true-positive mutations from the mutation catalog and 12,900 cross-patient negative calls based on the mutation catalog of the other patients. Out of the alteration catalog, 88 true mutations and 1330 cross-patient negative calls had a signal for the alteration.

Using this threshold, DREAMS-*vc* called 84.1% of the tumor mutations with signal, while Shearwater and Mutect2 called 70.5% and 67.0%, respectively (Table 2). F1 and G-mean scores were calculated to assess the performance of the models by using the cross-patient mutations as negative controls. G-mean is the geometric mean of sensitivity and specificity, and F1 is the harmonic mean of precision and sensitivity. For G-mean, DREAMS-*vc* performed better than Shearwater and Mutect2; however, the F1 score of Shearwater was very similar to DREAMS-*vc*, due to lower false-positive rate of shearwater (Table 2). Considering all mutations observed in the tumors, including those without signal in plasma, we found that about 38.7% could

**Table 2** Mutation calling performance (true-positive mutations and cross-patient negative calls)

| | Full alteration catalog[a] | | Catalog alterations with signal[b] | | | |
|---|---|---|---|---|---|---|
| | Sensitivity | Specificity | Sensitivity | Specificity | F1 | G-mean |
| DREAMS-*vc* | 0.387 | 0.997 | 0.841 | 0.955 | 0.479 | 0.896 |
| Shearwater | 0.325 | 0.998 | 0.705 | 0.963 | 0.461 | 0.824 |
| Mutect2 | 0.309 | 0.997 | 0.670 | 0.948 | 0.376 | 0.797 |

[a] Full alteration catalog consisting of $n = 191$ true-positive mutations, and $n = 12,900$ potential cross-patient negative calls
[b] Catalog of alterations with signal consisting of $n = 88$ true positive mutations, and $n = 1330$ potential cross-patient negative calls

be recalled in Pre-OP liquid biopsy samples. DREAMS is generally more sensitive than Mutect2 and Shearwater, especially for stage II cancer mutations. Higher stage cancer is expected to have a larger signal for cancer mutations. Mutations called by DREAMS-vc but missed by Shearwater generally have low allele frequencies (Additional file 1: Section S7).

By setting the threshold based on a 5% false-positive rate in the cross-patient mutation set, the validation mutation set can be used as negative controls. The true positives are still the same 191 mutations of which 88 has a signal for the alteration. The negatives are the 500 validation positions multiplied with the 87 tested samples, giving a total of 43,500 possible alterations of which 1920 had a signal. With this set, we obtained an 86.4% true positive rate, compared to 76.1% for Shearwater and 67.0% for Mutect2 (Table 3). DREAMS-*vc* scored highest in both F1 and G-mean scores. Here, DREAMS-*vc* performed distinctly better than Shearwater and Mutect2.

A common measure used to predict the presence of ctDNA is the estimated tumor fraction in plasma. DREAMS-*cc* combines the mutational evidence across the mutation catalog, to estimate the tumor fraction with an accompanying *p*-value for the presence of cancer (see the "Methods" section). We aimed to detect cancer in the Pre-OP samples, since cancer is present and should, in theory, be detectable given enough ctDNA is present in the blood. As a negative control, we attempted to detect cancer in each Pre-OP sample (Tested Sample) with the mutation catalog from all other patients (Candidate patient) (Fig. 6a). In case of shared mutations between the mutation catalogs, these were eliminated to prevent false positives. As a benchmark, we constructed a cancer call score using the product of the individual Bayes factors across the mutation catalog from Shearwater, resulting in a similar tendency (Fig. 6b), and for Mutect2, we used the highest scoring variant, since we do not have a natural way of combining the scores (Fig. 6c). The performance of calling cancer can be assessed by treating the cross-patient mutation catalogs as expected negatives and calculate an AUC score (Fig. 6d). Performance was compared using the $5 \times 2$ cross-validation setup as above (Fig. 4d). The AUC was similar between DREAMS-*cc* and Shearwater with respect to calling cancer; however, DREAMS-*cc* showed an increased performance. Mutect2 does not seem to be suited for cancer calling in this way. As for variant calling, we only included the samples with mutational signal to showcase and compare the performance of the different methods in discriminating tumor from error signal.

For the patients with stage I and II CRC, we found tumor-supporting reads in 47.5% (19/40) and 73% (33/45) of the Pre-OP samples, respectively. We called cancer in 36.6%

**Table 3** Mutation calling performance (true-positive mutations and validation alterations)

| | Full alteration catalog[a] | | Catalog alterations with signal[b] | | | |
|---|---|---|---|---|---|---|
| | Sensitivity | Specificity | Sensitivity | Specificity | F1 | G-mean |
| DREAMS-*vc* | 0.398 | 0.997 | 0.864 | 0.946 | 0.448 | 0.904 |
| Shearwater | 0.351 | 0.995 | 0.761 | 0.920 | 0.323 | 0.837 |
| Mutect2 | 0.309 | 0.997 | 0.670 | 0.952 | 0.388 | 0.799 |

[a] Whole catalog consisting of $n = 191$ true-positive mutations, and $n = 43,500$ validation alterations

[b] Catalog of positions with signal consisting of $n = 88$ true positive mutations, and $n = 1920$ validation alterations

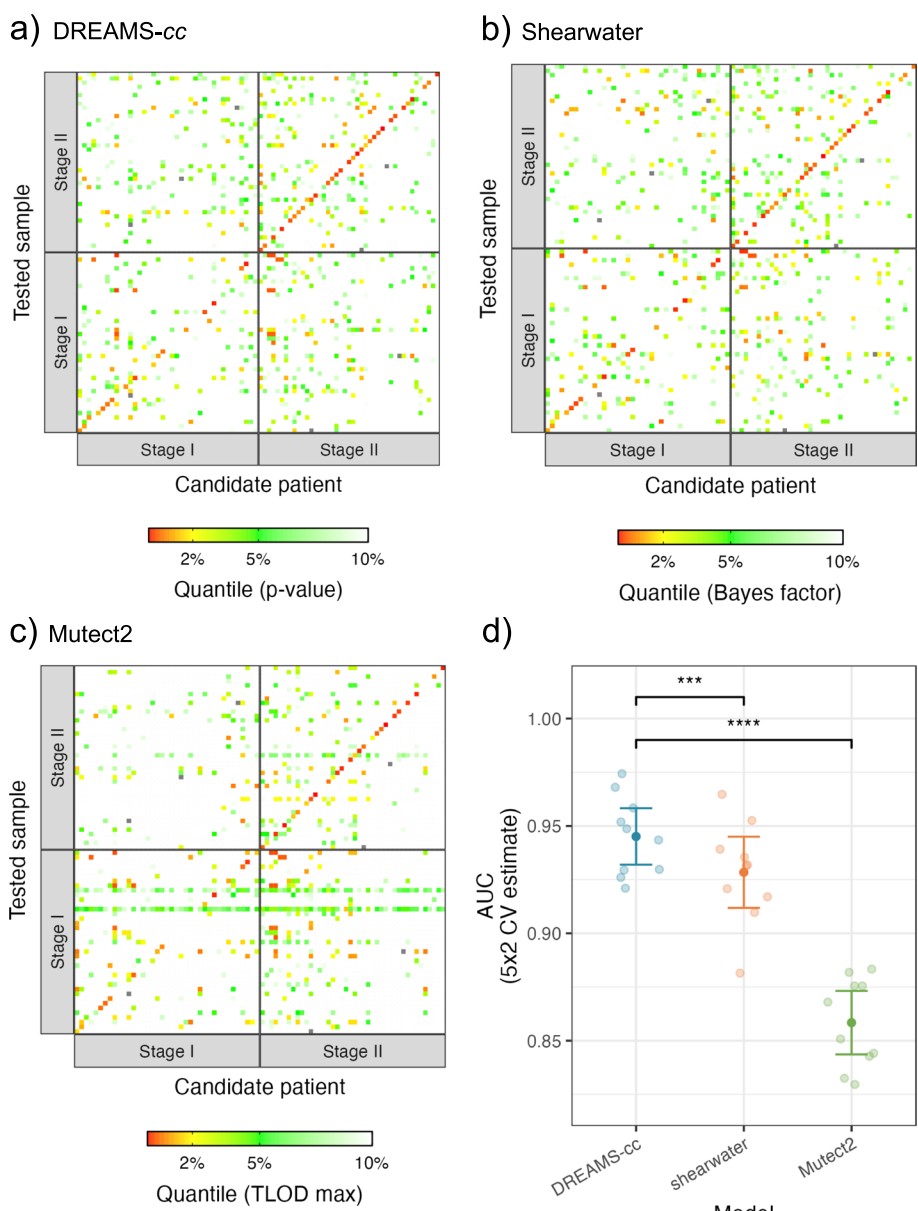

**Fig. 6** Prediction of cancer using **a** DREAMS-cc, **b** Shearwater, and **c** Mutect2. For each patient's LB-sample (*y*-axis), the mutation catalog (*x*-axis) for every candidate patient is used for calling cancer. The patients are stratified into patients with stage I and stage II CRC. The diagonal is showing the result of using a patient's own mutation catalog for cancer calling and constitutes the expected positives. The off-diagonal is the cross-patient results, for which the mutation catalog is filtered with the patient's tumor and germline variants prior to cancer calling, and thus these are expected to be negative. The color scheme is chosen based on the matched quantiles of the rank from the *p*-value, combined Bayes factors and max TLOD from **a** DREAMS-cc, **b** Shearwater, and **c** Mutect2, respectively. The cancer predictions show the results from one split in the 5 × 2 CV. **d** AUC performance of DREAMS-cc, shearwater, and Mutect2 with respect to calling cancer using the 5 × 2 CV

of the stage I CRC patients, corresponding to 78.9% (15/19) of the patients with a mutational signal. We called cancer in 72.1% of the stage II CRC patients, corresponding to 93.9% (31/33) of the patients with signal. These results were obtained while still limiting

the false-positive rate to 5% in cross-patient cancer calls with a non-zero mutational signal.

Detailed analysis of the false-positive cancer calls reveals that most are due to a specific KRAS G12V variant: chr12:25,245,350 C > A. This variant is common in colon cancer, and it is therefore not surprising to find in the patients [25]. However, the mutation was not found in the patient's corresponding tumor or buffycoat samples. A possible explanation for this is that the mutation is not detected in the tumor biopsy due to subclonality [26] or that there is an underlying germline signal that was not caught in the buffycoat.

### Discussion

We have developed DREAMS, as a new approach for modeling the error rates in sequencing data that incorporates information from both the local sequence context and read-level information. DREAMS is intended for settings that rely on accurate error identification and quantification. We applied the error model for low-frequency ctDNA variant calling (DREAMS-*vc*) and cancer detection (DREAMS-*cc*).

The error rate was found to vary depending on several of the proposed read-level features. Surprisingly, fragment size was found to be correlated with the error rate, with the smallest error-rates being observed for fragment sizes corresponding to the mononucleosomal and di-nucleosomal lengths (Fig. 3b). Fragments that deviate from these in length may have been degraded in the blood for a longer time and thereby accumulated more errors. Fragments of ctDNA are generally shorter and error rates are generally highest in short fragments, which shows the importance of accurate error modeling [27, 28]. The error rate was also found to vary with the strand, and symmetric mismatches occurred at different rates (Fig. 3e). The G > T/C > A asymmetry can be explained by the hybridization capture protocol only targeting one strand and thus only capturing oxidative damage of that strand [14]. A similar mechanism might explain the C > T/G > A asymmetry in the case of cytosine deamination. The error rate varied with the position in the read and was especially increased in the beginning of reads (Fig. 3d). This may be because the ends of fragments are prone to damage [14] and in thermodynamic equilibrium with being single stranded. The error rate also varied depending on whether the read was the first or second in the pair (Fig. 3d). Besides being intermitted by a PCR amplification step when sequencing, the reads differ in composition and length of adapters sequenced prior to the insert, which might cause this difference.

Training a background error model using DREAMS does not require known mutation sites in aligned reads (BAM-files), as it only aims to model the errors. For optimal ctDNA detection, the error model should describe the error distribution among normal cfDNA in the evaluated samples. Training data can originate from normal samples or potentially from mutation-filtered cancer samples, as in this study. Since error patterns are highly dependent on laboratory procedures, the same protocol should be used for training samples and subsequent testing samples. From the training samples, the common patterns of the errors occurring during library preparation, sequencing, etc., can be learned and would generalize to the testing samples. Error signals that are only found in a smaller subset of training samples, such as sample or batch effects, could lead to small biases. However, these should not have any significant effect on the final model,

if the model is trained across multiple samples gathered over time, thereby learning the general error patterns of the specific assay. Conversely, if the amount of data in a single sample is large, the error model could be trained on the sample itself, which would potentially yield a highly specific model that accounts for sample-specific error patterns. As shown here, even a single sample can provide sufficient data to get a relatively good performance (Additional file 1: Section S6), and similarly, it could potentially be advantageous to only train on same batch samples to diminish batch effects.

Since the biogenesis of ctDNA fragments may differ from normal cfDNA, they might have a different error profile, which could potentially lead to an error model not fitting normal cfDNA optimally. Large ctDNA contents in the training data might therefore affect ctDNA detection performance negatively and therefore best avoided. The samples here used for model training are expected to have low to no ctDNA content, and the performance presented does not indicate that our error model was affected by this.

The error model has been implemented using a neural network, allowing the feature set to be tailored to capture the relevant information of a specific setting. Analysis of the feature importance revealed that several of the proposed read-level features are useful in predicting the error rate in sequencing data (Fig. 4a). Most features presented in this paper are general to NGS data; however, not all sequencing protocols use UMI-based error correction, rendering UMI-related features redundant. In particular, UMI cannot be exploited for shallow whole-genome sequencing as read groups cannot be formed. In such cases error rates would be increased, making accurate error modeling as performed by DREAMS even more important.

Compared to simpler methods, the presented approach is more computationally demanding, due to training of the neural network model and the use of complex data extracted from BAM-files. A neural network is a simple and flexible approach for bridging the gap between a complex set of contexts and read-level features and the error rate of a given read position but might not be the most efficient solution. The model can be trained on a regular laptop within a few hours, which should only be done once, when the training dataset is defined. Using the trained model and the statistical modules adds no significant computation time for calling mutations and cancer in the current setting. However, very large mutation catalogs are expected to increase the computation time for DREAMS-*cc*.

DREAMS was built to exploit read-level features under the assumption that these affect the error rate in sequencing data. Thus, the power of this approach increases with the variability in the error rate explained by read-level features. Thereby, less emphasis is put on mismatches that are likely errors, and more confidence in the potential tumor signal from other mismatches. Conversely, if read-level features are not improving error prediction, the performance is expected to be similar to methods working with simpler summary data. Although DREAMS use information about the local sequence-context, strong regional effects on the error rate are not expected to be captured by the model. An added benefit of DREAMS being position agnostic is that it can be used to predict error rates for positions for which no training data is available. In principle, the method is fit for both deep sequencing of panels and shallow sequencing of whole genomes.

In all performance comparisons, DREAMS-*vc* performed better than the other methods in calling tumor mutations. This indicates that read-position level features can

improve performance in separating error from mutational signal. Similarly for cancer detection, DREAMS-*cc* performed better than Shearwater and Mutect2. Cancer was detected in most (73%) of stage II CRC cancer patients and a third (34%) of stage I patients. While Shearwater and Mutect2 could benefit from trimming the read ends, DREAMS can exploit the signal even in parts of the read with an increased error rate.

There are false-positive cancer and mutation calls, some of which could potentially be explained by clonal hematopoiesis of indeterminate potential (CHIP) or an unexpected error signal. To reduce the signal from CHIP, we have excluded positions with significant presence of non-reference nucleotides, found in the germline samples; however, a low signal might still be present. Remaining false-positive calls might be due to regional effects or sample-specific artifacts. Many of the false-positive mutation calls in the Pre-OP samples were found to be a mutation leading to the KRAS G12V variant, and it could therefore potentially be explained by a sub-clonal variant that was not identified in the tumor sample or a germline signal of clonal hematopoiesis of indeterminate potential (CHIP) that was not identified in the buffycoat samples. KRAS G12V is found to be an early driver event across multiple cancer types and therefore not expected to be particularly often subclonal [29]; however, it has been identified as a potential driver of clonal hematopoiesis [30].

Sensitive variant calling in liquid biopsies can provide non-invasive insight into tumor genetics, which can potentially enable personalized treatment of patients and be a cost-effective approach for cancer screening. DREAMS-*cc* integrates evidence across a mutation catalog to increase sensitivity in cancer detection. Cancer detection is expected to get more sensitive as the number of mutations in the catalog rises. A potential application of DREAMS-*cc* could be tumor-agnostic cancer detection based on a catalog of commonly known tumor variants.

The approach presented in this paper does not rely on tumor-specific signals, such as the fragment size distribution of fragmentations patterns of ctDNA, mutational signatures, and expression information, which differentiates it from seemingly similar methods such as INVAR [19] and MRDetect [18]. In the case of INVAR, the statistical model underlying the test for the presence of cancer uses the differences in fragment length distribution between normal cfDNA and ctDNA to help discriminate the fragments. Furthermore, INVAR uses a simple tri-nucleotide context error model compared to DREAMS' read-specific error model that uses multiple read-level features. Compared to MRDetect, we try to solve a similar problem using a neural network as the backbone, but with very different approaches. The Neural Network in MRDetect aims to discriminate between normal cfDNA and ctDNA by training on curated sequencing data. In DREAMS, we aim to predict the probability of an error occurring at a position in a read by training on bulk sequencing data from samples with minimal ctDNA concentration, or samples from healthy individuals. However, the performance of DREAMS could potentially be improved by incorporating a tumor-specific foreground model into the statistical framework to help guide if a fragment is indeed ctDNA, which would require a curated data from ctDNA fragments, which is rarely available. Additionally of cancer-specific regional properties or fragment behavioral information could potentially further increase sensitivity.

In this paper, we focus on the single nucleotide variants, but the model could be extended to be able to look for indels. The underlying ideas in DREAMS are not restricted to variant calling and could be used in other tasks of sequencing data analysis such as advanced error filtering.

## Conclusion

We have presented the DREAMS error rate model and demonstrated the importance of using read-level features for modeling the errors in NGS data. The model was validated in a tumor-informed setting, using DREAMS-*vc* for variant calling and DREAMS-*cc* for cancer detection in patients with CRC. DREAMS-*vc* allowed accurate detection of mutation signal in plasma samples extracted prior to curative intended surgery with improved performance compared to state-of-the-art methods. This highlights the importance of including read-level information in modeling the background error rate. Furthermore, DREAMS-*cc* demonstrated the ability to combine signal from multiple mutations known from the tumor biopsy for improved cancer detection. DREAMS-*cc* was able to call cancer in 73% of Pre-OP samples from CRC stage II patients and 34% of CRC stage I patients. Potential future applications of DREAMS include analysis of WGS data and tumor-agnostic cancer detection. The approach presented with DREAMS is generally applicable across NGS applications that need accurate handling and quantifications of errors, and the presented algorithms (DREAMS-*vc* and DREAMS-*cc*) are only examples of how to exploit this. The specific application presented in this paper is implemented as a user-friendly R package [31, 32].

## Methods

### Error rate prediction using read level information

In this study, we present a method called DREAMS (Deep Read-level Modelling of Sequencing-errors) for estimating the error rate at each read position using features of the individual read and the genomic context of the position. In practice, this is achieved by predicting the probability of observing each allele given the describing features of a position in a read and considering the probabilities of observing the alternative alleles as the error rates. The read-specific features can include information such as the read position, the strand of the mapped read, the length of the fragment, and UMI-group size. The read position refers to the cycle number at which the position was sequenced starting with the first nucleotide of the fragment, thus disregarding cycles used for reading primers, adapters, unaligned ends, etc. Context-specific features contain information about the genomic sequence surrounding the position, including the neighboring bases (tri-nucleotide context), the complexity, and GC-content. The local complexity is calculated as the Shannon entropy for both single nucleotides and pairs. Similarly, the local GC content is calculated as the fraction of C and G nucleotides. In principle, any feature that can be thought to affect the error rate of a read position can be added to improve the error rate prediction. Another possible feature would be the positional read quality score given by the sequencing machine. However, the estimated quality for the collapsed consensus reads were all capped at the same high value and thus excluded as they do not include any information for further modeling.

### Data

Data for a read position can be extracted from a read mapping (BAM-file) with sequencing data from a next-generation sequencing experiment. The training data for the model consists of a set of read positions from multiple samples, for which the observed allele is denoted together with the relevant features. This means that the training data includes both matches, where read positions where the observed allele is equal to the reference allele and mismatches where the observed and reference allele differ. Mismatches that correspond to known single nucleotide polymorphisms found in the germline samples are excluded from the training. Assuming that the training samples are non-cancerous means that all remaining mismatches in the dataset can be assumed to be errors that have occurred on a molecular level in the body or lab, or during sequencing of the sample.

The mismatches are extracted from the BAM-file using the mismatched positions annotated in the MD-tag. The equivalent genomic position is found, and the 11- and 3-mer context is extracted from the reference genome and used for calculation of local sequence-context features. The UMI errors and UMI count are extracted from the cE and cD tags generated by the CallMolecularConsensusReads from fgbio used for calling UMI consensus reads. Information about the insertions and deletions is extracted from the cigar tag. The fragment size is the insert-size (isize), and the read position is the position in the read sequence from the 5′-end of the read. Strand and first in pair are extracted from BAM flag where this information is encoded in a bitwise fashion.

The model assumes that the input data for both training and testing is based on readings of unique fragments, so each position in a fragment is only represented in one read. This can be assured using unique molecular identifiers (UMIs) and by trimming overlapping read positions in the read pairs.

As training on every single read position in every single read is very demanding and inefficient, we employ a methodology akin to importance sampling where we extract all the mismatches from the data and randomly sample a subset of the non-mismatches. To account for this skew induced by down-sampling one category of the training data, a rescaling scheme inspired by [33] is used on the predicted error rates. The method is outlined in Additional file 1: Section S8.

### Neural network model

#### *Structure of the neural network*

To predict the error rate at a given read position, we use a multilayer perceptron (MLP) which is a simple neural network setup with multiple fully connected layers. The neural network allows us to use the features without prior knowledge of how they interact among each other or how they affect the error rate. The neural network is trained using a set of read positions where the features describing the read positions are used as inputs and the observed allele as output.

For a given read position, the possible observed outcomes are the alleles A, T, C, or G. Interpreting this as a random event, the observed allele can be seen as an outcome from a four-dimensional multinomial distribution with one trial. Let $X_{ij}$ represents the observed allele in read $j$ at position $i$ and $D_{ij}$ be the set of observed features for that read

position. For a non-mutated, homozygote position, the observed allele should predominantly be the reference allele, and any observations of non-reference alleles would be considered errors. In this situation, $P(X_{ij} = A | D_{ij})$ would be close to 1 if $A$ was the reference allele for read position $(i,j)$, and $P(X_{ij} = x | D_{ij}), x \in \{T, C, G\}$ would be the error rates for the remaining three alleles. Given a set of observations $\{(x_{ij}, D_{ij})\}_{i=1}^{N}$, it is then possible to write the log-likelihood function for the observed data:

$$
\begin{aligned}
l\Big( &\big\{ (x_{ij}, D_{ij}) \big\}_{i,j} \Big) \\
&= \sum_{i,j} \log\big( P(X_{ij} = x_{ij} | D_{ij}) \big) \\
&= \sum_{i,j:x_{ij}=A} \log\big( P(X_{ij} = A | D_{ij}) \big) + \sum_{i,j:x_{ij}=T} \log\big( P(X_{ij} = T | D_{ij}) \big) + \\
&\quad \sum_{i,j:x_{ij}=C} \log\big( P(X_{ij} = C | D_{ij}) \big) + \sum_{i,j:x_{ij}=G} \log\big( P(X_{ij} = G | D_{ij}) \big)
\end{aligned}
$$

The problem now becomes how to estimate the distribution $P(X_{ij} | D_{ij})$ above. To do this, start by defining the probability functions via the SoftMax function:

$$
P(X_{ij} = a | D_{ij}) = \frac{e^{f_a(D_{ij})}}{\sum_{a' \in \{A,T,C,G\}} e^{f_{a'}(D_{ij})}}
$$

where $f_a(D_{ij})$ is a predictor function for the allele $a$ using the observed information $D_{ij}$. As an example, for classic multinomial logistic regression, a linear predictor function is chosen such that $f_a(X_i) = \beta_a \cdot X_i$, where $\beta_a$ is a vector of feature-specific weights that can be found by maximizing the log-likelihood function. To get a more flexible model, a neural network is chosen, since this can approximate any arbitrary predictor function well including arbitrary interactions between input features. To do this $P(X_{ij} = a | D_{ij})$ can be interpreted as the output from a neural network model where SoftMax is used as the last activation function and $f_a(D_{ij})$ is the output from the last hidden layer. To train such a model inspiration is drawn from likelihood theory and the negative log-likelihood function is chosen as the loss function to minimize.

### Architecture

The neural network model allows for high flexibility in the choice of features and requires very limited prior knowledge about the effect of the features on the error rate. The neural network was selected to be a MLP with an input layer, three hidden layers and an output layer. The dimension of the input layer depends on the selected input features, the hidden layers have a configuration of 128, 64, and 32 nodes with a ReLu activation function, and the output layer contains 4 nodes with SoftMax activation, as explained above, corresponding to probability of observing each of the 4 alleles. The configuration of hidden layers can be varied, depending on the input data and the available computational resources. The models were trained using the Keras library (2.3.0) in R, which is an interface that builds in Tensorflow (2.6.0) [34].

### Feature handling/embedding

The features are split into numeric, categorical, and embedded variables and handled accordingly. Categorical features are one-hot encoded, and the numeric features are

batch normalized. The trinucleotide context can be seen as the three distinct features: reference allele and the two neighboring bases. These can be handled as categorical features with individual one-hot encoded 4-dimensional inputs using 12 ($3 \times 4$) input nodes in total. Alternatively, a 64-dimensional ($4 \times 4 \times 4$) one-hot encoded input of the entire trinucleotide context (TNC) can be used. We will employ another alternative that takes the 64-dimensional feature in the input layer and embeds it into a continuous 3-dimensional vector before including it in the model alongside the remaining input features. Thereby, the model can learn the relationship between the contexts, and cluster contexts that have a similar effect on the error rate close together and vice versa.

### Assessing cancer status across a catalog of multiple mutation candidates

Based on the neural network error model developed above, it can now be assumed that the individual error rates for a given position in each read are known. In this section, the error rates will be exploited to develop a statistical framework for estimating the tumor fraction in a sample based on a catalog of candidate mutations. This framework can ignore some mutation candidates if these are not found in the sample, for example due to relatively low allelic frequency due to sub-clonality in the tumor or due to little tumor in the circulation. Reduction in the candidate mutations allows for a comprehensive mutation catalog to be used, where mutation candidates with limited evidence may be excluded. The subset of candidate mutations is selected statistically by finding mutations with a consistently high mutational signal, and the tumor fraction is estimated based on these candidates. This subset of mutations is then used in a statistical procedure for testing if the observed mutational signal exceeds what we would expect if no mutated DNA were present, making it possible to determine the cancer status of a patient based on the sample.

### The statistical model

Start by introducing $Z_i$ as a variable that controls the presence of a given mutation on the site $i$, such that $Z_i = 1$ represent the case where the site is mutated, and $Z_i = 0$ when it is not. Furthermore let:

$$Z_i \sim Bernulli(r)$$

Thus, given a catalog of possible mutations, $r$ is the probability that each of them is present in the sample. For site $i$ let $R$ be the germline reference allele and $M$ the alternative allele of interest. Furthermore, it is assumed that the germline site is homozygote, such that any signal from non-reference alleles must be due to errors or mutational signal from a tumor. To model the molecular composition of the fragments covering site $i$, let $Y_{ij} \in \{R, M\}$ be the true error-free nucleotide of the $j$'th fragment. If the $i$'th mutation is not present in the sample ($Z_i = 0$), we are sure that the true nucleotide of the fragment is the reference and thus the following distribution holds:

$$P(Y_{ij} = R | Z_i = 0) = 1, \quad P(Y_{ij} = M | Z_i = 0) = 0$$

To model the mutational DNA present in the sample, let $f > 0$ denote the tumor fraction. This is the fraction of the DNA in the blood that originates from tumor cells. Assuming that the mutation of interest is (sufficiently) clonal in the tumor, i.e., half of

the DNA in the tumor has this mutation, the probability of a given fragment having the mutation is $f/2$. Using this, the following distribution for $Y_{ij}$ can be assumed when the mutation is present in the sample ($Z_i = 1$):

$$P(Y_{ij} = R|Z_i = 1) = 1 - \frac{f}{2}, \quad P(Y_{ij} = M|Z_i = 1) = \frac{f}{2}$$

To model the errors that occur in NGS data, let $X_{ij}$ be the observed nucleotide in fragment $j$ at position $i$. Assume that the distribution of $X_{ij}$ depends only on the corresponding true nucleotide $Y_{ij}$, in the sense that the event $X_{ij} \neq Y_{ij}$ corresponds to the observation being an error. This distribution is exactly what the neural network model described above aims to approximate using the observed features $D_{ij}$. To simplify notation, the dependence of $X_{ij}$ on $D_{ij}$ will be omitted from the notation in the following. Note that observations $X_{ij}$ outside $\{R, M\}$ will have little information about the true nucleotide $Y_{ij}$. Furthermore, since the error rates generally are low, the difference between including interactions between all four possible alleles and only the two allele of interest is negligible. Thus, to simplify the following calculations, we assume that $X_{ij} \in \{R, M\}$. In practice, this means that all fragments, $j'$, for which $x_{ij'} \notin \{R, M\}$ are eliminated from the analysis. Using this assumption, we define the probability of observing the alternative allele in a reference allele position as the following error rate:

$$e_{ij}^{R \to M} = P(X_{ij} = M|Y_{ij} = R, X_{ij} \in \{R, M\}) = \frac{P(X_{ij} = M|Y_{ij} = R)}{P(X_{ij} = R|Y_{ij} = R) + P(X_{ij} = M|Y_{ij} = R)}$$

Conversely, for a fragment that stems from a tumor cell and contains the mutated allele, we define:

$$e_{ij}^{M \to R} = \frac{P(X_{ij} = R|Y_{ij} = M)}{P(X_{ij} = R|Y_{ij} = M) + P(X_{ij} = M|Y_{ij} = M)}$$

### Estimating the tumor fraction and mutation presence

In this section, we will develop a procedure for estimating the tumor fraction ($f$) and mutation presence probability ($r$). For this, let $i \in \{1, \ldots, K\}$ be the index of a catalog of $K$ candidate mutations, $N_i$ the corresponding number of covering reads and $\left\{ (x_{ij})_{j \in \{1,\ldots,N_i\}} \right\}_{i \in \{1,\ldots,K\}}$ all the observed alleles. First, we write the likelihood function for $f$ and $r$:

$$L\left(f, r \mid \left\{(x_{ij})\right\}_{i \in \{1,\dots,K\}, j \in \{1,\dots,N_i\}}\right)$$

$$= \prod_{i=1}^{K} P(Z_i = 0).$$
$$\prod_{j:x_{ij}=R} \left[P(X_{ij} = R \mid Y_{ij} = R)P(Y_{ij} = R \mid Z_{ij} = 0) + P(X_{ij} = R \mid Y_{ij} = M)P(Y_{ij} = M \mid Z_{ij} = 0)\right].$$
$$\prod_{j:x_{ij}=M} \left[P(X_{ij} = M \mid Y_{ij} = R)P(Y_{ij} = R \mid Z_{ij} = 0) + P(X_{ij} = M \mid Y_{ij} = M)P(Y_{ij} = M \mid Z_{ij} = 0)\right] +$$
$$P(Z_i = 1).$$
$$\prod_{j:x_{ij}=R} \left[P(X_{ij} = R \mid Y_{ij} = R)P(Y_{ij} = R \mid Z_{ij} = 1) + P(X_{ij} = R \mid Y_{ij} = M)P(Y_{ij} = M \mid Z_{ij} = 1)\right].$$
$$\prod_{j:x_{ij}=M} \left[P(X_{ij} = M \mid Y_{ij} = R)P(Y_{ij} = R \mid Z_{ij} = 1) + P(X_{ij} = M \mid Y_{ij} = M)P(Y_{ij} = M \mid Z_{ij} = 1)\right]$$

$$= \prod_{i=1}^{K} (1 - r) \cdot \prod_{j:x_{ij}=R} \left(1 - e_{ij}^{R \to M}\right) \cdot \prod_{j:x_{ij}=M} e_{ij}^{R \to M} +$$
$$r \cdot \prod_{j:x_{ij}=R} \left[\left(1 - e_{ij}^{R \to M}\right) \cdot \left(1 - \frac{f}{2}\right) + e_{ij}^{M \to R} \cdot \frac{f}{2}\right] \cdot \prod_{j:x_{ij}=M} \left[e_{ij}^{R \to M} \cdot \left(1 - \frac{f}{2}\right) + \left(1 - e_{ij}^{M \to R}\right) \cdot \frac{f}{2}\right]$$

Getting a maximum likelihood estimate (MLE) of $f$ and $r$ by optimizing this expression analytically is not tractable. However, by seeing $Y_{ij}$ and $Z_i$ as latent variables, estimates can be found by employing an EM-algorithm (Additional file 1: Section S9). For now, assume that $\widehat{f}$ and $\widehat{r}$ are MLEs of $f$ and $r$ respectively.

To test if a sample has a significant content of mutational DNA, we focus on the parameters in the model. By representing the hypothesis of a negative sample as a tumor fraction of 0 and no mutations present ($H_0 : f, r = 0$) and a positive sample as a positive tumor fraction and some mutations present $\left(H_A : f > 0, r \geq \frac{1}{K}\right)$, a likelihood ratio test can be used to test for significance. Note that $r \geq \frac{1}{K}$ in $H_A$ corresponds to at least one mutation being present in the sample. The LR-test statistic for this test is:

$$Q = -2\log \frac{L\left(0, 0 \mid \left\{(x_{ij})\right\}_{i \in \{1,\dots,K\}, j \in \{1,\dots,N_i\}}\right)}{L\left(\widehat{f}, \widehat{r} \mid \left\{(x_{ij})\right\}_{i \in \{1,\dots,K\}, j \in \{1,\dots,N_i\}}\right)}$$

Since there are 2 free parameters in the model, it can be assumed that $Q$ is approximately $\chi^2(2)$-distributed, and a *p*-value can be obtained as follows:

$$p_{val} = 1 - F_{\chi^2(2)}(Q)$$

Using this statistical model for cancer calling on top of the error rate predictions from DREAMS, we refer to it as the DREAMS-cc.

### Calling individual mutations

In the special case where the number of mutations in the catalog is $K = 1$, the algorithm outlined above can be thought of as a regular variant caller. In this case, the concept of some mutations not being present in the sample is unnecessary, as the presence of the single mutations of interest can be governed solely by the tumor fraction $f$. The algorithm above is easily modified to handle this by assuming that $r = 1$, and using one degree of freedom for the $\chi^2$-distribution in the significance test. The equations in the EM-algorithm can also be simplified by making this assumption. We refer to the variant caller as DREAMS-*vc*.

## Supplementary Information

> **Additional file 1.** Supplementary material, Supplementary methods, tables, and figures [12, 16, 24, 33, 35–43].
>
> **Additional file 2.** Review history.

### Acknowledgements

We thank the participating CRC patients and the Danish Cancer Biobank for contributing clinical material. We also thank the IMPROVE study group for patient inclusion: Kåre Andersson Gotschalck (Horsens Hospital), Lene Hjerrild Iversen (Aarhus University Hospital), Uffe Schou Løve (Viborg hospital), Anders Husted Madsen (Herning Hospital), Ole Thorlacius-Ussing (Aalborg University Hospital), Ismail Gögenur (Køge Hospital), Per Vadgaard Andersen (Odense University Hospital), Jakob Lykke (Herlev Hospital), Peter Bondeven (Randers Hospital), and Nis Hallundbæk Schlesinger (Bispebjerg Hospital).

### Peer review information

Anahita Bishop and Andrew Cosgrove were the primary editors of this article and managed its editorial process and peer review in collaboration with the rest of the editorial team.

### Review history

The review history is available as Additional file 2.

### Authors' contributions

MHC, SD, CLA, and JSP conceived and designed the study. MHC and SD developed the statistical methods and the software under supervision by JSP with input from MHR and CLA. MHR, AF, IL, CD, JN, KAG, and LHI acquired patient samples and generated patient data, including NGS data. MHC, SD, MHR, AF, CLA, and JSP analyzed and interpreted the patient data. SD and MHC wrote the article under the supervision of CLA and JSP with revisions and suggestions from the other authors. All authors read and approved the final manuscript.

### Authors' Twitter handles

Amanda Frydendahl, @amanda_fbj; Iben Lyskjær, @illyskjaer; Jesper Nors, @JesperNors; Lene H. Iversen, @LeneHjerrild; Claus L. Andersen, @ClausLindbjerg; Jakob Skou Pedersen, @skouped.

### Funding

MHR, AF, LI, JN, and CLA were funded by Aarhus University, Lundbeck Foundation (R180-2014–3998), Dansk Kræftforsknings Fond (FID1839672), Innovationfund Denmark (9068-00046B), Danish Cancer Society (R133-A8520-00-S41, R146-A9466-16-S2, R231-A13845, R257-A14700), NEYE foundation, Frimodt-Heinke Foundation, and Novo Nordisk Foundation (NNF17OC0025052). SD, MHC, and JSP were funded by Aarhus University, the Independent Research Fund Denmark | Medical Sciences (8021-00419B), the Danish Cancer Society (R307-A17932), and Aarhus University Research Foundation (AUFF-E-2020–6-14).

### Availability of data and materials

The data that support the findings of this study are available on request from the corresponding author [CLA]. The data are not publicly available due to them containing information that could compromise the research participants' privacy. Access to the data requires that the Danish National Committee on Health Research Ethics ethically approve the requestors' intended use of the data and that the legal entity of the data requestor enters into a data protection agreement with the Danish data controller, the Central Denmark Region. DREAMS is available as an R software package Zenodo and GitHub under GPL-3.0 license [31, 32].

## Declarations

### Ethics approval and consent to participate

The Committees on Biomedical Research Ethics in the Central Region of Denmark have approved the study (J. No. 1–10-72–3-18). The study was performed in accordance with the Declaration of Helsinki and all participants provided written informed consent.

### Consent for publication

Not applicable.

### Competing interests

The authors declare that they have no competing interests.

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

## 

