## [**Additional file 2.** Review history. · Genome Biology]

Review History

First round of review

Reviewer 1

Were you able to assess all statistics in the manuscript, including the appropriateness of statistical tests used? Yes, they are sound and appropriate.

Comments to author:

The manuscript reported an innovative method to increase the sensitivity of detecting rare mutations in ctDNA by accurately modeling read-level errors. The method provided a general framework to model the errors in consensus reads constructed from sequencing reads with UMI. This led to a more refined approach to making use of UMIs in sequencing error reduction and control. Some concerns can be addressed to further strengthen the manuscript.

Major:

How to train a background error model is the key for the proposed algorithm. Starting in Line 288, some general discussion was given about the requirements and choices of training data. It may be helpful to expand the paragraph to include some limiting factors. For example, normal samples may have different patterns in fragment ends than the tumor samples. How would this impact the error model? Random errors can happen during library preparation. Training over multiple samples would lead to a more general model. However, fluctuation in library prep and sequencing may lead to uncommon errors that would not be picked by the model. In the sentence started in Line 293, a sample specific model was proposed without supporting results. How effective would it be? How large the data must be?

The error model improved the performance of DREAMS-vc predominantly among the stage II CRC patients (Figure 5b). How this performance gain was influenced by mutant allele frequency (i.e., mutational signal/fraction)? In another word, the error model appeared to offer more critical information for a specific range of mutant allele frequency (MAF). Please include the MAF of those mutations detected by DREAMS-vc but missed by mutect2 or shearwater.

A higher error rate in cfDNA fragment ends is well known. Figure 3d again confirmed this. Caller accuracy can be improved by excluding the first 12 bases of each read. Have you tested this simple approach for mutect2 and shearwater? How will their improved performance compare to that of DREAMS-vc?

There might be an interaction between fragment length and UMI family size. It would be interesting to look into this.

For future work, it will be important to demonstrate the utilities of DREAMS-vc by applying it to some sequencing datasets generated from reference ctDNA samples, for example, those made publicly available by the SEQC2 consortium.

Minor:

Wrong reference format in Line 47-48.

Line 546 & 548, N should be N_i . Some N's were also missing the subscript in Supplementary Section 6.

Line 578-579, please correct the wrongly structured sentence.

Reviewer 2

Were you able to assess all statistics in the manuscript, including the appropriateness of statistical tests used? Yes

Comments to author:

The authors present a new read-based error correction model for cfDNA mutation profiling and analysis. Identification of cancer mutations in cfDNA is currently hampered by cfDNA error rates often dominating low variant allele fractions, leading to poor accuracy. This work is therefore very timely and the read-level approach (DREAMS) is novel. The analysis reveals several interesting read-level features that determine error rates in cfDNA. A (justified) limitation of the model is that it needs to be retrained/fitted on new cohorts/datasets to account for experiment and protocol specific bias. The performance of the model is favourable when compared to existing approaches (Mutect2, Shearwater) for variant calling. Overall, this study and new method should be of broad interest to the cfDNA research field. Below are my main points that could further strengthen the manuscript and its conclusions.

Major points

Lack of comparison with iDES and MRDedge/detect

In the introduction, I feel iDES is incorrectly presented as a method that aggregates mutation data for ctDNA detection. iDES, as presented in the original paper, is "serial application of molecular barcoding and background polishing, integrated digital error suppression (iDES)." This idea is conceptually similar to DREAMS. Why have the authors not compared their error correction performance with iDES? Please justify or add iDES to comparison. Similarly, MRDedge/detect (Zviran et al.) also includes a read-level computational error suppression approach. Would it be possible to compare DREAMS with this approach?

Lack of comparison with state-of-the-art variant caller, Strelka2

Strelka2 (Kim et al., Nature Methods, 2018) is considered a gold-standard somatic mutation calling algorithm, often outperforming Mutect2 in benchmarks. The authors would strengthen their benchmark if they could add Strelka2 to the variant calling comparison.

Minor points

Discussion lack comparison with related approaches

The discussion is lacking a section where the method is compared to existing approaches that also adopt some form of error correction (Invar / MRDEdge / MRDdetect). The authors should discuss how their work and model compares to these existing approaches.

Only demonstrated use case is targeted sequencing data

I appreciate that DREAMS is a flexible model that could potentially be used for either targeted, Exome or WGS data. It would therefore significantly strengthen the paper if the authors could demonstrate how the method performs with Exome (like Invar) or WGS data (like MRDdetect) data. Perhaps some of the same samples from their pre/post-OP cohort could be sequenced with WGS?

P. 7, l. 131

The authors should discuss the key assumptions when creating a training dataset this way. Could some of the remaining variants still be real mutations, e.g. sub clonal variants or CHIP? And would this pose a problem?

P. 9, l. 187

Variant callers often have default thresholds for the minimum number of reads required to support a mutation call. I assume the authors have ensured that this threshold is configured to allow a variant caller like Mutect to call a mutation with just 1 variant read.

P. 12

Why was Mutect not considered alongside Shearwater for the cancer detection problem?

P. 12, and discussion, on KRAS G12V mutations

It's interesting that the authors identify a large number of potential false-positive mutations at KRAS G12V. The authors speculate that this could be related to CHIP, sub clonal alterations, or germline alterations. Since this is a common (potential) artefact, this part could be further strengthened with additional data. Is KRAS G12V frequently sub clonal in colorectal tumors (e.g. TCGA data)? Are G12V mutations often found in germline databases or CHIP studies (e.g. Ptashkin 2018)?

Reviewer #1:

The manuscript reported an innovative method to increase the sensitivity of detecting rare mutations in ctDNA by accurately modeling read-level errors. The method provided a general framework to model the errors in consensus reads constructed from sequencing reads with UMI. This led to a more refined approach to making use of UMIs in sequencing error reduction and control. Some concerns can be addressed to further strengthen the manuscript.

We thank the reviewer for acknowledging the novelty and applicability of our approach. We appreciate the comments to the manuscript, and we hope that you also find that the changes have strengthened the paper.

Major:

How to train a background error model is the key for the proposed algorithm. Starting in Line 288, some general discussion was given about the requirements and choices of training data. It may be helpful to expand the paragraph to include some limiting factors.

For example, normal samples may have different patterns in fragment ends than the tumor samples. How would this impact the error model? Random errors can happen during library preparation. Training over multiple samples would lead to a more general model. However, fluctuation in library prep and sequencing may lead to uncommon errors that would not be picked by the model.

We agree that the proposed applications are dependent on the learned error model and hence on how it is trained. We have expanded our discussion of this aspect and included additional evaluation of the effect of the number of samples used for training.

For cfDNA, reads stemming from tumor cells may have different error profiles than reads stemming from blood cells or other normal cells, given different fragment generation processes. In principle such differences could aid the detection of ctDNA. To facilitate this, it is therefore attractive to train the error model on cfDNA samples with no or low levels of ctDNA. By training on post-OP samples, we aim to fulfill this. We have now included the following discussion of this aspect [Line 319-320 + Line 334-339]:

“For optimal ctDNA detection, the error model should describe the error distribution among normal cfDNA in the evaluated samples. [...] Since the biogenesis of ctDNA fragments may differ from normal cfDNA, they might have a different error profile, which could potentially lead to an error model not fitting normal cfDNA optimally. Large ctDNA contents in the training data might therefore affect ctDNA detection performance negatively and therefore best avoided. The samples here used for model training are expected to have low to no ctDNA content, and the performance presented does not indicate that our error model was affected by this.”

The errors in the library preparation and sequencing are some of the errors that we aim to model. There might be variations in the error rates across samples and batches, therefore it is advisable to select training data as close to the test data as possible. This could potentially be achieved by sample specific models or models trained on the same batch, as mentioned.

We cannot evaluate such approaches with our current data (see below), but have included additional discussion:

The following Discussion has been extended (Line 323-328):

“From the training samples, the common patterns of the errors occurring during library preparation, sequencing etc. can be learned, and would generalize to the testing samples. Error signals that are only found in a smaller subset of training samples, such as sample or batch effects could lead to small biases. However, these should not have any significant effect on the final model, if the model is trained across multiple samples gathered over time, thereby learning the error patterns specific to the assay.”

In the sentence started in Line 293, a sample specific model was proposed without supporting results. How effective would it be? How large the data must be?

The sample specific model has the potential of making very specific error models. However, this will require more data from each sample. We have evaluated the dependence on the size of the training data and found that performance increases with the number of samples. However, even models trained on a single sample can perform similar to Shearwater. This shows that even with sparse data, we can train a relatively good model.

The following figure has been added to Supplementary Section 6 and the performance across the training data sizes is briefly discussed in the Results section and the Discussion:

Results [Line 199-202]:

*“In addition, we evaluated how the performance of DREAMS depended on the size of the training set and found it increased with the number of training samples, with the largest gain seen when going from one to two samples (**Supplementary Section 6**).”*

Discussion [Line 330-333]:

*“As shown here, even a single sample can provide sufficient data to get a relatively good performance (**Supplementary Section 6**), and similarly it could potentially be advantageous to only train on same batch samples to diminish batch effects. ”*

The error model improved the performance of DREAMS-vc predominantly among the stage II CRC patients (Figure 5b). How this performance gain was influenced by mutant allele frequency (i.e., mutational signal/fraction)? In another word, the error model appeared to offer more critical information for a specific range of mutant allele frequency (MAF). Please include the MAF of those mutations detected by DREAMS-vc but missed by mutect2 or shearwater.

To evaluate the range of MAFs where DREAMS shows a detection advantage, the MAF of variants called by DREAMS-vc and Shearwater were compared. As shown in the figure below, DREAMS is generally more sensitive to calling mutations in Stage II cancer, as the reviewer suggests.

Generally, the variants called using DREAMS-vc and not by Shearwater were of low frequency (Median MAF of 0.001 vs median MAF of 0.002 for all common calls). For stage II the difference is higher, (Median MAF of 0.001 vs median MAF of 0.003 for all common calls). From this analysis it appears that the improved sensitivity of DREAMS is based on improved calling of low frequency variants in Stage II CRC patients. The analysis has been added to Supplementary Section 7, and the following has been added to the Result section [Line 244-248]:

“DREAMS is generally more sensitive than Mutect2 and Shearwater, especially for Stage II cancer mutations. Higher stage cancer is expected to have a larger signal for cancer mutations. Mutations called by DREAMS-vc but missed by Shearwater generally have low allele frequencies. (Supplementary Section 7)”

A higher error rate in cfDNA fragment ends is well known. Figure 3d again confirmed this. Caller accuracy can be improved by excluding the first 12 bases of each read. Have you tested this simple approach for mutect2 and shearwater? How will their improved performance compare to that of DREAMS-vc?

We thank the reviewer for opening the discussion of end trimming of reads. In the initial submission we trimmed two bases from the beginning of reads and one base was removed from the end of reads. This was due to an apparent increase in noise at these positions.

Since DREAMS is the only one of the compared methods that take read position into account, trimming should be less beneficial for DREAMS, and could, as the reviewer notice, improve the relative performance of Shearwater and Mutect2.

To assess the effect of trimming we compared different trimmings (start:end). Generally, the performance of DREAMS-vc is decreased by trimming, especially in the beginning of reads in the current data. Indicating that DREAMS models the noise sufficiently to be able to exploit the signal in the start and end of reads. The performance of Shearwater is slightly improved with trimming of 2 bases at the read start. The performance of mutect2 can also be improved with some trimming.

To illustrate the power of DREAMS, it should naturally be presented without hard data filtering such as read trimming, and the manuscript results have therefore been updated with untrimmed results.

A discussion of trimming has been added to the Result section [Line 220-228]:

*“The explorative feature analysis indicated an increased error rate at the beginning and end of reads (**Figure 3d**). DREAMS takes the read position into account and can thereby compensate for the increased error rate at these positions. Other methods such as Mutect2 and Shearwater are not aware of read ends and the performance of these can potentially improve by trimming these. To investigate the effect of trimming, we evaluated the performance of each method when trimming 0, 2, 6, or 12 of the bases in the beginning of reads or 0, 1, or 2 of the bases in the end of reads (**Figure 5b**). We found that Shearwater can improve performance by trimming 2 bases from the beginning of reads and Mutect2 can improve by trimming the last base of each read. For DREAMS the performance is only decreased when trimming, especially in the beginning of reads.”*

And Discussion [Line 370-371]:

“While Shearwater and Mutect2 could benefit from trimming the read ends, DREAMS can exploit the signal even in parts of the read with an increased error rate.”

Following figure has been added to Figure 5:

There might be an interaction between fragment length and UMI family size. It would be interesting to look into this.

There could be interactions between the fragment length and the UMI family size. If present, the Neural Network as parameterized would be able to learn these.

To investigate if there is any correlation/interaction between fragment size and UMI group size in the exploration data (see density plot below), we permuted the UMI group size randomly, to see if it changes the distribution. That does not seem to be the case, which indicates that the features vary independently/are uncoupled.

For future work, it will be important to demonstrate the utilities of DREAMS-vc by applying it to some sequencing datasets generated from reference ctDNA samples, for example, those made publicly available by the SEQC2 consortium.

We agree with this. However, we have tried to locate read-level data from the SEQC2 consortium but have unfortunately only been able to locate VCF-level data at this point.

Minor:

Wrong reference format in Line 47-48.

Fixed

Line 546 & 548, N should be Ni. Some N's were also missing the subscript in Supplementary Section 6.

Fixed

Line 578-579, please correct the wrongly structured sentence.

Fixed

Reviewer #2:

The authors present a new read-based error correction model for cfDNA mutation profiling and analysis. Identification of cancer mutations in cfDNA is currently hampered by cfDNA error rates often dominating low variant allele fractions, leading to poor accuracy. This work is therefore very timely and the read-level approach (DREAMS) is novel.

The analysis reveals several interesting read-level features that determine error rates in cfDNA. A (justified) limitation of the model is that it needs to be retrained/fitted on new cohorts/datasets to account for experiment and protocol specific bias.

The performance of the model is favourable when compared to existing approaches (Mutect2, Shearwater) for variant calling. Overall, this study and new method should be of broad interest to the cfDNA research field. Below are my main points that could further strengthen the manuscript and its conclusions.

We thank the reviewer for acknowledging the novelty and the scope of the method that we have developed.

Major:

Lack of comparison with iDES and MRDedge/detect

In the introduction, I feel iDES is incorrectly presented as a method that aggregates mutation data for ctDNA detection. iDES, as presented in the original paper, is "serial application of molecular barcoding and background polishing, integrated digital error suppression (iDES)." This idea is conceptually similar to DREAMS. Why have the authors not compared their error correction performance with iDES? Please justify or add iDES to comparison.

iDES is indeed miscategorized as a method that aggregates mutation signal. This has been corrected and the following has been added to describe iDES [Line 75-78]:

"Another approach, iDES[12], finds mutations in paired reads by combining a specialized stranded barcoding scheme and a polisher that aims to filter out erroneous mutational signal based on a number of criteria, including an error model. Mutations are then called based on the remaining variant signal."

iDES is a method that is very focused on a barcoding scheme and it is therefore not generally applicable. The polishing aspect of iDES can to some degree be applied, however without information about the barcoding. The polishing is most similar to Shearwater, since it involves a position-wise error model, however instead of being used for scoring variants the error model is used for polishing the signal – blacklisting positions. After polishing the variants can be ranked based on the allele frequency, which is a relatively simple form of variant calling.

We have performed this aspect of their implementation; however, it is not a really a fair comparison to iDES as we are not using the required barcoding scheme during library preparation. The performance of iDES is slightly worse than Mutect2 and Shearwater in this setup.

Since iDES can't be compared directly, we have added it to Supplementary Section 5.

Similarly, MRDedge/detect (Zviran et al.) also includes a read-level computational error suppression approach. Would it be possible to compare DREAMS with this approach?

MRDedge/detect is indeed comparable, as they also use read-level features. A major difference is that MRDedge/detect are foreground models that are trained to classify something as tumor-reads/not-tumor-reads, whereas DREAMS is a background model focused on learning the errors.

As mentioned in the Background section, a foreground model requires curated data from known ctDNA fragments, whereas a background models (such as iDES, Shearwater and DREAMS) only need error signal (control samples) to train on. MRDedge is mostly (if not only) used for whole genome sequencing (WGS) data and in that setting it could in principle be possibly to compare with DREAMS, if appropriate training data as required by MRDedge was available. However, such a WGS comparison is beyond the scope of this study.

Lack of comparison with state-of-the-art variant caller, Strelka2

Strelka2 (Kim et al., Nature Methods, 2018) is considered a gold-standard somatic mutation calling algorithm, often outperforming Mutect2 in benchmarks. The authors would strengthen their benchmark if they could add Strelka2 to the variant calling comparison.

We recognize that Strelka2 is a commonly used method for mutation calling. One drawback for comparison with Strelka2 is that it cannot be “forced” to score variants that it doesn’t recognize as mutations with sufficient signal, however it can give an overall score for each position. Comparing Strelka2 with scores for each position or only the positions that it recognizes gives the performance shown in the performance figure above. It generally does not seem to perform very well in this setup.

Since Strelka2 can’t be compared directly, we have added it to Supplementary Section 5.

The following has been added about Strelka2 in the Background section:

“Similar to Mutect2, Strelka2 realigns the reads and then uses a statistical model to determine the likelihood of a variation being real by analyzing base quality, read mapping quality and depth at each position.”

And to reference the Supplementary Section in the Result section [Line 198-199]:

*“DREAMS was also compared to iDES[12] and Strelka2[16], however, these methods are not directly comparable and are therefore presented in **Supplementary Section 5.**”*

Minor:

Discussion lack comparison with related approaches

The discussion is lacking a section where the method is compared to existing approaches that also adopt some form of error correction (Invar / MRDEdge / MRDdetect). The authors should discuss how their work and model compares to these existing approaches.

We have now included some additional discussion of how our proposed methods compare to Invar and MRDEdge/MRDetect. Though Invar includes a tri-nucleotide-based error model, the main difference is that the DREAMS applications exploit read-level features and not any tumor-specific (foreground) features. DREAMS could potentially be expanded with a foreground model in the future. The major limitation for this is the availability of curated tumor mutation signal from ctDNA. The following discussion has been added (Line 389-398):

“ The approach presented in this paper does not rely in tumor specific signals, such as the fragment size distribution of fragmentations patterns of ctDNA, mutational signatures, expression information, etc., which differentiates it from seemingly similar methods such as INVAR [19] and MRDetect [18]. In the case of INVAR, the statistical model underlying the test for the presence of cancer uses the differences in fragment length distribution between normal cfDNA and ctDNA to help discriminate the fragments. Furthermore, INVAR uses a simple tri-nucleotide context error model compared to DREAMS’ read specific error model that uses multiple read level features. Compared to MRDetect we try to solve a similar problem using a neural network as the backbone, but with very different approaches. The Neural Network in MRDetect aims to discriminate between normal cfDNA and ctDNA by training on curated sequencing data. “

Only demonstrated use case is targeted sequencing data

I appreciate that DREAMS is a flexible model that could potentially be used for either targeted, Exome or WGS data. It would therefore significantly strengthen the paper if the authors could demonstrate how the method performs with Exome (like Invar) or WGS data (like MRDdetect) data. Perhaps some of the same samples from their pre/post-OP cohort could be sequenced with WGS?

It would be very interesting to compare the performance with methods suited for WES or WGS data. However, we find that a proper performance comparison at the exome or WGS level is a large endeavor, which would require more time and in-depth analysis than permitted within the realm of this study. Additionally, to compare against the state-of-the-art WGS methods, such as MRDdetect, we would require curated read-level ctDNA (foreground) data.

P. 7, l. 131

The authors should discuss the key assumptions when creating a training dataset this way. Could some of the remaining variants still be real mutations, e.g. sub clonal variants or CHIP? And would this pose a problem?

We agree that our training setup could in principle include small amount signal from real mutations or CHIP. However, filtering steps taken as a precautionary measure should

minimize this problem, as supported by performance. We have added the following to elaborate the assumptions for the training data to the Discussion (Line 185-189):

“The tumor variant positions were excluded from the training data. Potential signals from sub-clonal variants that are not detected in the tumor should be infrequent and present at low levels, and therefore have minimal effect on the error model. By excluding variants found in the germline samples, we expect to reduce the potential signal of clonal hematopoiesis of indetermined potential (CHIP).”

P. 9, l. 187

Variant callers often have default thresholds for the minimum number of reads required to support a mutation call. I assume the authors have ensured that this threshold is configured to allow a variant caller like Mutect to call a mutation with just 1 variant read.

Yes, methods like Mutect2 have a lot of filters and usually only returns the calls with some relatively high confidence. We have force-called using Mutect2 and thereby let Mutect2 score all of the variants that we compare across, including cases supported by just one variant read, which we can then rank based on the TLOD score. This is now made more explicit in the Result section (Line 195-196):

“All positions with at least one observed mismatch were scored with each method and included in the performance calculations (Figure 5a).”

P. 12

Why was Mutect not considered alongside Shearwater for the cancer detection problem?

We agree that it would be natural to include Mutect2 for the cancer detection problem as well. Mutect2 was not considered for cancer detection since we don't have a natural way of aggregating the evidence, which we do with Shearwater and DREAMS. One simple way to make the comparison is to take the position with the highest variant signal (best scoring variant) of the given mutation catalog. We have now included the result of this analysis in Figure 6 and in the Result sections [Line 274-284].

P. 12, and discussion, on KRAS G12V mutations

It's interesting that the authors identify a large number of potential false-positive mutations at KRAS G12V. The authors speculate that this could be related to CHIP, sub clonal alterations, or germline alterations. Since this is a common (potential) artefact, this part could be further strengthened with additional data.

Is KRAS G12V frequently sub clonal in colorectal tumors (e.g. TCGA data)?

We have investigated the subclonality of the KRAS G12V in the WES data of an independent cohort of 340 colon cancer tumor samples (unpublished, patient sensitive clinical data). We do not see any indication that the KRAS G12V mutations should be subclonal more often than

other frequent driver mutations in the calls. In the PCAWG study of cancers' evolutionary histories they found that KRAS mutations to be particularly an early event and hence we do not expect it to be especially often subclonal.

Are G12V mutations often found in germline databases or CHIP studies (e.g. Ptashkin 2018)?

We have looked in the InteGen database of clonal hematopoiesis drivers and found that it is a known CHIP driver mutation, and could therefore be a CHIP signal. The following has been added to the Discussion section:

“KRAS G12V is found to be an early driver event across multiple cancer types, and therefore not expected to particularly often subclonal[29], however, it has been identified as a potential driver of clonal hematopoiesis[30].

Second round of review

Reviewer 1

The revised manuscript has addressed all questions raised for the earlier version.

Please see this data descriptor paper (<https://www.nature.com/articles/s41597-022-01276-8>) for more information about the read-level data from the SEQC2 consortium.

Reviewer 2

The authors have addressed all my concerns, the additional comparisons with other approaches further highlight the accuracy and sensitive of the method.